# Asbestos Waste Treatment—An Effective Process to Selectively Recover Gold and Other Nonferrous Metals

**Omirserik Baigenzhenov** [1], **Alibek Khabiyev** [1,*], **Brajendra Mishra** [2], **Indira Aimbetova** [3], **Sultan Yulusov** [1], **Inkar Temirgali** [1], **Yerzhan Kuldeyev** [1] **and Zhanar Korganbayeva** [4]

1    Mining-Metallurgy Institute, Satbayev University, 22a Satpaev Str., Almaty 050013, Kazakhstan
2    Metal Processing Institute, Worcester Polytechnic Institute, 100 Institute Road, Worcester, MA 01609-2280, USA
3    Faculty of Natural Sciences, Akhmet Yassawi International Kazakh-Turkish University, 29 B. Sattarkhanov Ave., Turkestan 161200, Kazakhstan
4    Institute of Natural Sciences and Geography, Abai University, 13. Dostyk Ave., Almaty 050010, Kazakhstan
*    Correspondence: a.khabiyev@satbayev.university

**Abstract:** We investigated the potential of tailings generated from chrysotile asbestos fiber production as a source of iron, nonferrous metals, and gold. We proposed the use of granulometric separation and systematically examined different enrichment processes, namely magnetic separation, gravity concentration, and enrichment using a Knelson concentrator, to extract the valuable components. The characterization of the initial tailing samples revealed that it comprises primarily of serpentine, brucite, antigorite, hematite, vustite, sillimanite, and magnesium oxide. Using the suggested enrichment process, we isolated gold, chromite, and nickel-cobalt concentrates as valuable products in addition to magnetite. The new approach exhibited high separation efficiency for iron, nonferrous metals, and gold, allowing their satisfactory extraction.

**Keywords:** asbestos tailings; complex processing; enrichment; nonferrous metals; gold





## 1. Introduction

In Kazakhstan, the recycling of tailings is an important issue, as this refuse occupies huge areas and generates environmental pollution. In the 65 years of Zhetikara chrysotile asbestos deposit exploitation (based in the city of Zhetikara, Kostanay region), the local processing plant processed about 310 million tons of asbestos ores. The output of marketable asbestos was only about 6–7% [1], and the remaining tailings (amounting to over 300 million tons) were transferred to refuse dumps that occupy hundreds of hectares of land. Several studies have established the negative impact of refuse on the ecological environment [2–6].

The ores of the Zhetikara geological–industrial chrysotile asbestos deposits are serpentinized peridotites, dunites, and serpentinites. These rocks are known for their high content of elements such as magnesium, iron, platinum, chromium, cobalt, and nickel. The asbestos-bearing serpentinites of the deposits can also be enriched with gold, as serpentinites in contact with each other may undergo partial listvenitization and develop dykes of more recent granitoids in the rift zones [1,6].

Industries have conducted activities related to the reuse of tailings for some time. Previous studies established that the least extensively treated refuse is suitable for ballasting railways, as a filler for asphalt road paving, and as a coarse-grained topping for tar-and-gravel roofing, etc. Furthermore, hundreds of thousands of tons of washing refuse have been utilized annually for these purposes. Researchers have developed and pilot-tested a technology for producing magnesium from the serpentinites contained in asbestos refuse. Extraction from serpentinites is the cheapest method of producing magnesium.

In 2010, the world produced 2010 thousand metric tons of asbestos (Figure 1). While these statistics show a decreasing tendency since 2021, about 1200 thousand metric tons of asbestos is still stored worldwide [2].

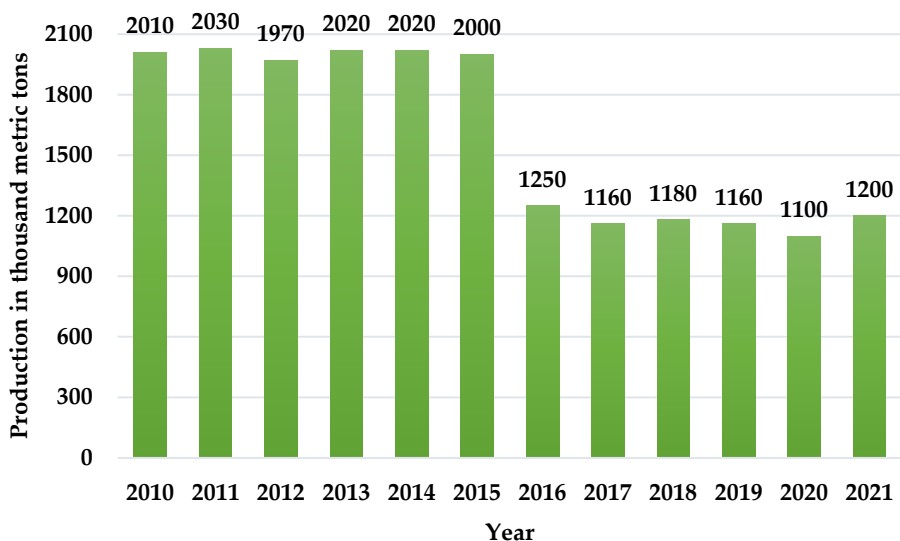

**Figure 1.** Mine production of asbestos worldwide from 2010 to 2021 [2].

In 2018, the world consumed a total of about 1180 thousand tons of asbestos (Figure 2) of which 18.3% was consumed in Kazakhstan (United States Geological Survey (USGS) data) [2].

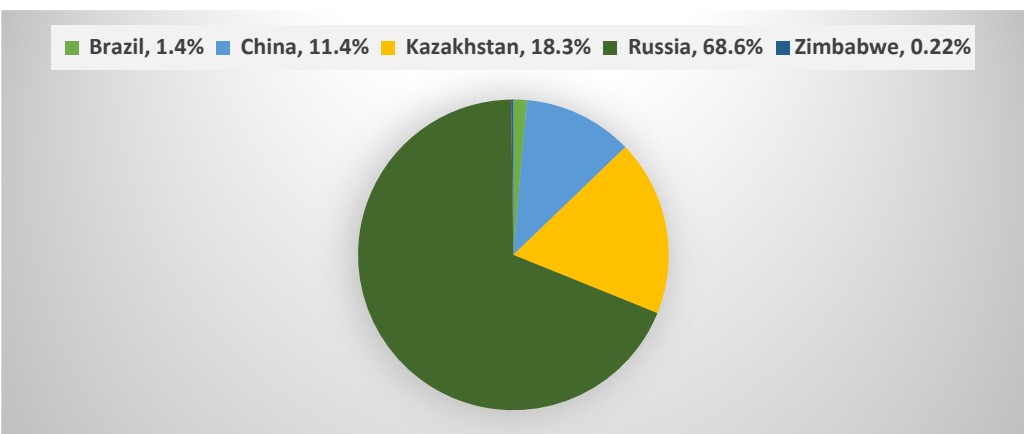

**Figure 2.** Percentages of global asbestos consumption in 2018 [2].

From 1975 to 1977, G.M. Teterev supervised a project aimed at making industrial use of the asbestos refuse from the Zhetikara deposit, considering the possibility of extracting iron, nickel, cobalt, chromium, and other components from the magnetite and nickel–cobalt flotation concentrates [6]. Since the asbestos production waste in Kazakhstan contains significant amounts of Ni, Fe, and Au, and the abovementioned methods cannot achieve their complete recovery, we developed a technological scheme that eliminates the main disadvantages of the previously developed technologies.

We established a recycling pathway with the implementation of mineral and metallurgical processing technologies that incorporated a fundamentally new method of mineral processing based on gravity enrichment, enhancing the extraction of metals and improving the environment. The extraction of nonferrous metals from asbestos production waste became a reality relatively recently, and the related technology is constantly improving. The interest in this area is due to the fact that 1) the serpentinite supply is almost inexhaustible and is readily available for processing, and 2) the cost of the raw materials is very low [7–11].

The principal difference between our study and previous studies is that through studying in detail the technological process of enriching asbestos ores, we concluded that

a new technogenic multicomponent deposit was being formed in its course. Asbestos extraction processes subject the ore to multiple crushing steps and transportation, which create the conditions for the gravitational enrichment of parts of the refuse with heavier valuable components.

Under our methodological plan, we aimed to develop a technology for processing tailings using small-scale laboratory equipment that would resolve the following issues:

- Ensuring the profitability of the processing method, which is possible if the following three targets are met: the elimination of expensive "fine" crushing steps from the technological process, instead using only the simplest and cheapest methods of enrichment; the expansion of the range of produced marketable products; and the separation of the most expensive groups of noble and nonferrous metals.
- Realizing the least harmful (from an ecological point of view) method of processing possible by avoiding the use of active chemical reagents that are harmful to human health and the environment (acids, alkalis, etc.).

## 2. Experiment

### 2.1. Materials

Asbestos tailings were sampled from an asbestos plant in the city of Zhetikara in the North Kazakhstan region. Samples were fine powder and did not need to be treated further before enrichment processes.

### 2.2. Methods

Instrumental Analyses

The average granulometric composition of asbestos production tailings found by the quartering method was determined by classification with analytical sieves. Previously, the material was dried. Humidity was determined by heating the initial industrial product in a vacuum drying oven at a pressure of 7.8 kPa and a temperature of 95–100 °C until a constant weight of the suspension was established. A measuring cylinder was used to determine the poured and tap bulk density. All the necessary weighing of samples during the work was carried out on an analytical scale of RA214C (Ohaus-Pioneer) with an error of 0.0005 g.

The elemental composition of asbestos tailings and enrichment products was determined using X-ray fluorescence analysis via an X-ray fluorescence wave dispersion combined spectrometer from Axios "PANalyical".

The mineralogical composition of asbestos tailings and enrichment products was determined by the usage of X-ray phase analysis via the D8 Advance X-ray diffractometer (BRUKER). Micrographs of the initial samples of asbestos tailings were obtained with a JEOL JXA-8230 scanning electron probe microanalyzer.

## 3. Experimental Flowsheet

As shown in Figure 3, the experimental process included a granulometric separation of the initial crushed ore, a separation of hydrocyclones, and an enrichment of magnetic separators, gravity concentration, and a separation at the Knelson concentrator.

We conducted experiments on enlarged laboratory samples (weighing up to 100 kg) using laboratory equipment in a sequence that simulated the continuous technological process of a similar high-productivity industrial apparatus, i.e., we established the methods for processing the tailings.

We obtained the small samples through the continuous application of pneumatic and oscillating screens, griddles, bypasses, and conveyors. In the case of manual sampling, we extracted the sample material from the lower part of the moving bed, since the heavy valuable ore minerals tend to concentrate in the lower half of the jigged bed of the crushed ore.

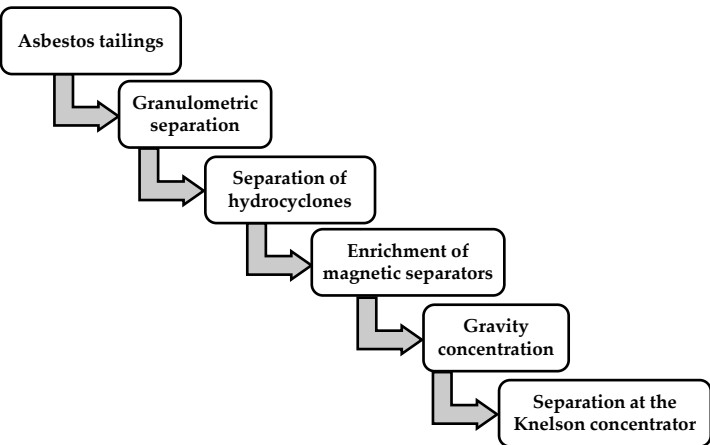

**Figure 3.** Schematic of experimental flowsheet.

The enlarged samples represented only the <0.25 mm class of the layer. The heavy minerals are concentrated in the lower, heavier layer of the pneumovibrating apparatus, 60–70% of which comprises heavy minerals (magnetite, chromospinelide, sulfides, olivine, pyroxene, etc.) attached to light minerals (serpentine, chrysotile asbestos, nemolite, brucite, carbonate, etc.). Only 30–40% of the minerals remain free from any attachments (due to previous comminution processes) and, as they take the form of separate grains, they can be separated from each other by their physical properties (magnetic, gravitational, etc.).

Most of the unenriched attachments comprised grains with a size greater than 0.25 mm and could thus be easily separated from the enriched part of the individualized mineral grains via a 0.25 mm sieve. Therefore, in contrast to the 12 small samples, the enlarged samples only contained materials smaller than 0.25 mm from the lower layer of the conveyors dumping tailings into the refuse dumps.

Based on our findings, we developed a scheme for the full technological process of tailing enrichment and identified several valuable industrial products: magnetite and chromite concentrates, gold, nickel and cobalt sulfides, and olivine and diopside sands.

*Description of processes for the separation of asbestos tailings components under laboratory conditions.* Figure 3 shows the recommended processes presented in the schematic technological scheme found during a separation process of the components of asbestos tailings and obtained in the process of enrichment of the chrysotile-asbestos ore. These processes include a granulometric separation of tails, a separation of tails on hydrocyclones, a magnetic separation of tails, a gravitational separation on the concentration table, and an enrichment at the Knelson concentrator.

*Granulometric separation of tailings.* Vibration screening is one of the most important and widely used technological operations in disintegration and plays a special role in energy conservation [12,13]. By itself, it is low-energy intensive and makes it possible to implement a fundamental principle—"do not crush anything superfluous"—which is a direct way to save energy costs for crushing and grinding mineral raw materials and other materials.

To optimize the control of screening crushing processes, it is necessary to use parameters that reflect the efficiency of all processes both separately and as a whole. Therefore, a class of −0.25 mm is proposed as a product of the screening process. The choice of this size range is since iron, nonferrous, and precious metals are generally more concentrated in the fine fraction of screening.

This process was selected as the first because in the process of screening asbestos tailings, iron, chromium, and precious metals can all be separated from serpentinite during the first screening stage.

*Separation of tailings on hydrocyclones.* At processing plants, hydrocyclones are traditionally used to classify crushed material, which are an inexpensive apparatus with a modest production footprint. The acting force of the hydrocycloning process is the centrifugal force

that arises in the flow of liquid supplied to the hydrocyclone through a branch pipe with a tangential inlet [14]. Under the influence of centrifugal forces, particles of a given size and density are released from the flow and are diverted down the hydrocyclone into its conical part, after which they enter the assembly hopper. Due to the high efficiency and simplicity of the device, its compact size, cheapness, and ease of operation, hydrocyclones are widely used in both phase separation processes and in the enrichment of various types of ores [15,16].

Hydrocycloning of the fraction of class −0.25 after screening enables the further division of the remaining amount of serpentine in the concentrate and increases the overall content of iron, chromium, and precious metals.

*Magnetic separation of tailings.* Many works have demonstrated the possibility of implementing the stadial separation of the finished concentrate during magnetic separation in a weak field. The results of studies of magnetic separation in a weak field of various enrichment products have shown that the greatest efficiency of separation of magnetite from accretions with rock is achieved at a lower magnetic field strength, in the range of 15.9–23.9 kA/m [17]. When the magnetic field strength decreases from 79.6 to 15.9 kA/m, the mass fraction of iron in the magnetic product increases to that of the conditioned product. It follows from this that a LIMS in a constant magnetic field with a voltage in the range of 15.9–23.9 kA/m can be used for the stadial production of conditioned concentrates [18,19].

The first stage of magnetic separation of tailing (HIMS) is carried out at a high magnetic field strength and serves to separate magnetic grains from non-magnetic grains.

At high-intensive magnetic separation (HIMS), the uncovered ore grains are separated from the coalescence of ore grains with the waste rock.

A two-stage magnetic separation process (HIMS and LIMS) allows the uncovered ore grains to be separated from the concentrate obtained after hydrocycloning. Moreover, during the process of separation at low intensive magnetic separation (LIMS), chromites are separated from the non-magnetic fraction that contains both gold and silicates.

*Concentration table.* Gravity enrichment via concentration tables is one of the most important ways of separating minerals, and in many cases, is the most expensive of all existing methods [20,21]. The principles of gravity separation are widely used for the direct enrichment of various ores and materials, as well as in ore preparation processing. This method allows the enrichment of ores and materials of a wide range of sizes—from 50 mm to 0.001 mm. Rectangular classical concentration tables are mainly used for enrichment, which are referred to as gravity-centrifugal apparatuses [22].

*Enrichment at the Knelson concentrator.* In the field of gravity enrichment, one of the most important problems is the creation of a high-performance apparatus for the efficient extraction of fine heavy mineral grains from ores and, in particular, free particles of precious metals. Over recent years, significant progress has been made in this direction and is associated with the emergence of a large number of valuable separators of various types, of which Knelson concentrators have the greatest distribution [23–25].

The principle of operation of the concentrator is based on the separation of incoming material in a centrifugal field into two fractions: "heavy" and "light." The separation of the material into fractions occurs as a result of the interactions among the washing water flow, centrifugal forces, and the gravity field that acts on the particle in a horizontally or obliquely rotating rotor.

The Knelson concentrator allows for the separation of silicates and for the recovery of a gold concentrate.

## 4. Results and Discussion

### 4.1. Results of Phase and Physico-Chemical Analysis of the Initial Tails

The average composition of these tailings contained: MgO (39.0–42.0%); $SiO_2$ (37.0–41.0%); CaO (1.1–1.6%); $Fe_2O_3$ (1.9–5.4%); FeO (1.0–2.7%); $Al_2O_3$ (0.8–1.4%); NiO (0.1–0.25%); $Cr_2O_3$ (0.1–0.25%); and Au (0.3 g/t). The mineral base of the ore was serpentine, chemically expressed as $3MgO·2SiO_2·2H_2O$ (Table 1).

**Table 1.** Phase composition of the initial asbestos tailings.

| Component | Formula | Weight, % |
|---|---|---|
| Serpentine | $Mg_3(Si_2O_5)(OH)_4$ | 53 |
| Brucite | $Mg(OH)_2$ | 6 |
| Magnesium–nickel oxides | $MgNiO_2$ | 2 |
| Antigorite | $Mg_6(Si_4O_{10})(OH)_8$ | 23 |
| Magnesium oxide | $MgO$ | 4 |
| Silimanite | $Al_2O_3 \cdot SiO_2$ | 3 |
| Hematite | $Fe_2O_3$ | 5 |
| Vustite | $FeO$ | 2 |

The humidity of the initial sample was 8.2%. The poured bulk density was 1.65 g/cm$^3$ and the tap bulk density was 1.97 g/cm$^3$. After drying to a constant mass, representative tailings samples were analyzed using both physical and chemical methods.

The results of the elemental analysis of the initial tails are presented in Table 2.

**Table 2.** Elemental composition of the main components of the initial asbestos tailings, %.

| Mg | Si | Al | Fe | Ca | Ni | Cr | Au, g/t |
|---|---|---|---|---|---|---|---|
| 23.4 | 17.8 | 0.42 | 5.3 | 0.34 | 0.19 | 0.15 | 0.3 |

We could not establish the forms of the Cr, Ni, and Fe contained in the tailings by X-ray diffraction analysis. Before the experiments, we dried the asbestos tailings at a temperature of 95–100 °C.

The X-ray analysis (Figure 4) showed that the sample comprised serpentinite and contained silica in the crystalline and amorphous states. These results were in line with the particle size analysis. We also observed particles larger than 10 microns in the cross-section of the sample, although we found it difficult to perform an EDX analysis on the individual particles due to interference from neighboring particles. However, the EDX analysis allowed us to identify several phases, such as serpentine, antigorite, and brucite, which confirmed the results of the XRD analysis. However, we could not identify gold in the asbestos tailings due to its low overall concentration.

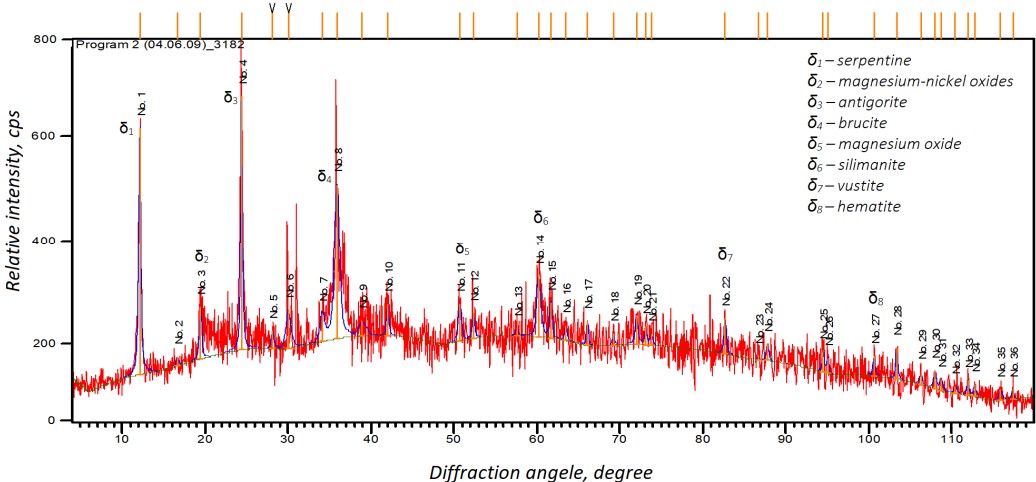

**Figure 4.** X-ray diffraction analysis of the initial asbestos products.

We used scanning electron microscopy (SEM) to study the morphology of the asbestos tailing samples. The SEM pictures showed that the asbestos tailings comprised a very fine material and that aggregates were formed (Figure 5).

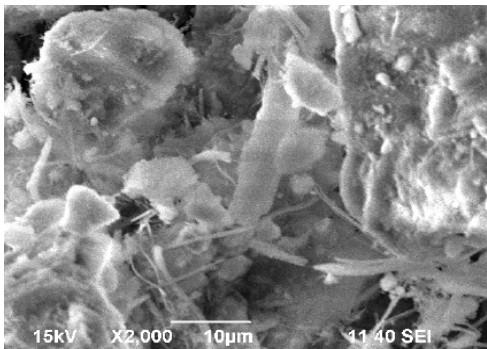

**Figure 5.** Scanning electron microscopy image of initial asbestos tailing.

*4.2. Results of Granulometric Analysis of the Initial Tails*

To study the possible distribution of ore components by size, the initial sample was fractionated into different fractions, from 0.50 to 2.00, from 0.25 to 0.50, and a fraction less than 0.25 mm, and samples of all fractions were analyzed. After a chemical analysis of the different fractions, it was found that magnesium silicates, being a fibrous material, were mainly concentrated in a large fraction; however, nonferrous metals and iron were concentrated in a small fraction (Table 3).

**Table 3.** Granulometric composition of asbestos tailings.

| Fraction Size, mm | Class Output, % | Content of Components, % | | | | | | | |
|---|---|---|---|---|---|---|---|---|---|
| | | MgO | $SiO_2$ | $Al_2O_3$ | CaO | $Fe_3O_4$ | FeO | NiO | $Cr_2O_3$ |
| from 0.50 to 2.00 | 39 | 40.1 | 38.3 | 0.72 | 0.46 | 2.18 | 1.1 | 0.11 | 0.10 |
| from 0.25 to 0.50 | 33 | 39.7 | 38.9 | 0.82 | 0.49 | 2.87 | 1.6 | 0.23 | 0.14 |
| <0.25 | 28 | 39.0 | 36.1 | 0.85 | 0.44 | 5.2 | 2.1 | 0.25 | 0.23 |

As noted in Section 3, representative samples were selected for laboratory experiments on tailings processing, and samples weighing 100 kg were selected for these experiments. Figure 6 shows a flowchart of the developed technological chain of tailing enrichment in the laboratory scale. It includes granulometric, gravitational, magnetic, and electromagnetic separation.

The data revealed in Figure 6 were presented in Table 4, which was supplemented with calculations for scaling and setting up a production flow for processing tailings of a 3.2-million-ton chrysotile-asbestos plant. A plant with an output of 3.2 million tons of chrysotile-asbestos tailings exists in northern Kazakhstan [1,9].

**Table 4.** The masses of products obtained on a laboratory scale and calculated during the processing of chrysotile-asbestos tailings with a volume of 3.2 million tons.

| Name of the Product | Mass of the Product | |
|---|---|---|
| | Obtained in the Laboratory Scale with a Volume of 100 kg of Chrysotile-Asbestos Tailings | Calculated for the Annual Yield of Chrysotile-Asbestos Tailings with a Volume of 3.2 Million Tons |
| Magnetite concentrate | 0.6 kg | 19.2 thousand tons |
| Chromite concentrate | 0.138 kg | 4.41 thousand tons |
| Nickel–cobalt concentrate | 0.01 kg | 1.895 thousand tons |
| Diopside sand | 1.06 kg | 34 thousand tons |
| Olivine forsterite sand | 1.59 kg | 50.9 thousand tons |
| Gold concentrate | 9.5 g | 5 tons |

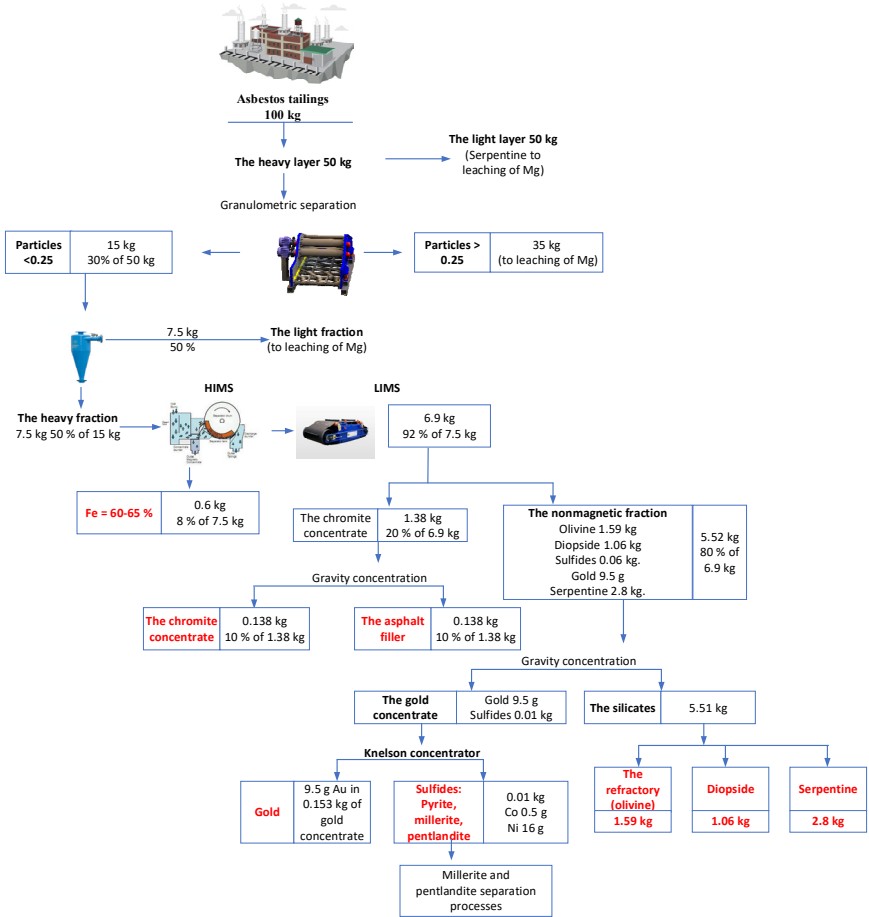

**Figure 6.** Flowchart of asbestos tailing processing.

Our estimations for the main stages of the technological chain and the industrial products generated are based on the annual dump-tailing output of the Zhetikara ore-processing plant, equal to 100 kg.

(1) Undersizing the heavy residue with a 0.25 mm sieve obtained a <0.25 mm yield of 30%, i.e., the partition process (separation) needed to be applied to 15 kg of tailings (see Figure 6). The analytical investigation showed that the concentration of gold in such a volume was 0.7 g/t (9.5 g), of which 50% was presumably in the form of gold nuggets; the rest comprised sulfides and magnetites.

(2) The hydrocyclonic gravitational separation of the original heavy tailings using a 0.25 mm sieve isolated two products in approximately equal parts. The heavy residue with a mineral specific gravity of over 3.5 g/cm$^3$ (collective gravity concentrate amounting to 50%) totaled about 7.5 kg. The float of roughly the same weight was sent to be leached for magnesium. Eighty percent or more of the heavy fraction comprised individualized minerals, which allowed for their subsequent separation into monomineralic products (concentrates) according to their physical properties.

(3) The LIMS and HIMS of the collective heavy gravitational concentrate obtained two products: (a) 0.6 kg of magnetite concentrate for cast iron and steel smelting (with a yield amounting to 8% of the initial volume); and (b) a nonferrous fraction (92% of the initial volume) of about 6.9 kg, which should subsequently be subjected to LIMS. The magnetite content in the magnetite concentrate was 82–90%, and the iron content was 60–65%, which is sufficient for iron production.

(4) The separation by LIMS of the 6.9 kg nonferrous fraction of the refuse, carried out via separators with a strong magnetic field, produced the following:

- A weak magnetic fraction, comprising a marketable chromite concentrate (0.138 kg, 10% of 1.38 kg) used to produce ferrochromium and chromite salts and an asphalt filler (0.138 kg, 10% of 1.38 kg).
- A nonmagnetic fraction of 5.52 kg (80% of the initial volume), comprising approximately 29% heavy silicates (olivine, forsterite, spinel, pyroxenes—1.59 kg); 51% light silicates (serpentine, 2.8 kg); 19.25% diopside (molding sands—1.06); small quantities (up to 1%) of nickel sulfides (millerite, pentlandite, pyrrhotite, etc.—0.06 kg); and gold (9.5 g). Chromite concentrate consists mainly of monomineral chromite with an insignificant admixture of pyrrhotite, magnetite, and serpentine. In the concentrate, the magnetite and serpentine are in the form of attachments.

(5) The gravity concentration of the 5.52 kg of nonmagnetic fractions allowed the production of the following half-products: gold concentrate (0.153 kg) and silicate concentrate (5.51 kg). Both products should be sent for "honing".

The developed scheme obtained the following industrial products: magnetite iron ore concentrate with an average iron content of 62–65%; chromite concentrate with an average chromium(III) oxide content of about 35–40%; gold nuggets of high fractionality (922–996 ppm) in the form of gold concentrate with a metal content of more than 50%; and a sulfide concentrate with a millerite–pentlandite–pyrite–pyrrhotite composition containing more than 16% nickel, more than 0.5% cobalt, 62 g/t gold, and olivine forsterite and spinel pyroxene sands that meet the requirements of the refractory, molding, and abrasive industries.

Based on our calculations, the annual Zhetikara chrysotile asbestos deposit tailings output could yield the following quantities of industrial products:

- Magnetite concentrate, suitable for iron production—0.6 kg;
- Marketable chromite concentrate—0.138 kg;
- Nickel–cobalt concentrate in the form of sulfides—0.01 kg;
- Diopside sand—1.06 kg;
- Olivine forsterite sand—1.59 kg;
- Gold concentrate—9.5 g.

## 5. Conclusions

This paper presents the results of a unique study on the enrichment of chrysotile asbestos tailings. The technology of their processing is examined with respect to the production of magnetite concentrate, chromite concentrate, nickel–cobalt concentrate, diopside sand, olivine forsterite sand, and gold concentrate.

Since the representative samples of chrysotile-asbestos tailings were from the Zhetikara deposit, and their average annual output in Northern Kazakhstan is 3.2 million tons, the possibility of creating a workshop for their processing at the existing enterprise was examined. The presented technological scheme of processing includes modern separation methods, such as a granulometric separation of tailings, a separation of tailings on hydrocyclones, a magnetic separation of tailings, a gravitational separation on the concentration table, and an enrichment at the Knelson concentrator, all of which make it possible to obtain concentrates of varying composition with a high yield. The application of our results can significantly increase the profitability of the development of asbestos deposits and increase the range of products produced. The described technology would also reduce environmental problems in the regions that surround the existing plant. We have tested the technology on a laboratory scale and recommend its implementation in the existing production.

The construction of a chrysotile-asbestos tailings processing plant using the proposed waste processing technology can pay for itself, due to profits from the production of gold concentrate within two to three years. Moreover, since additional substances such as strong acids and bases will not be used in this technology, large capital investments are not required for specialized tailings processing equipment.

**Author Contributions:** Conceptualization, O.B.; methodology, A.K.; validation, I.T.; formal analysis, S.Y.; investigation, O.B.; data curation, A.K.; writing—original draft preparation, I.A. and B.M.; writing—review and editing, Y.K.; visualization, Z.K.; supervision, O.B. and A.K. All authors have read and agreed to the published version of the manuscript.

**Funding:** This research was funded by the Science Committee of the Ministry of Science and Higher Education of the Republic of Kazakhstan (grant No. AP13268858).

**Data Availability Statement:** The data presented in this study are available on request from the corresponding author.

**Conflicts of Interest:** The authors declare no conflict of interest.

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
