# Peer review of "Asbestos Waste Treatment—An Effective Process to Selectively Recover Gold and Other Nonferrous Metals"

_recycling, doi:10.3390/recycling7060085_

Round 1

Reviewer 1 Report

The authors proposed a technical roadmap to extract valuable concentrates through sequentially physical processing, which is a good topic on the waste management of asbestos tailings. However, the manuscript lacks an essential description of material characterization and separation methods on the lab scale, and the presentation of quantitative outcomes of physical parameters, like density, size distribution, elemental compositions, and mineral phase compositions. Besides, presenting your experimental separation outcomes based on your estimation instead of real practice would be inappropriate.

Author Response

Response to Reviewer 1. Comments

Comments and Suggestions for Authors

The authors proposed a technical roadmap to extract valuable concentrates through sequentially physical processing, which is a good topic on the waste management of asbestos tailings. However, the manuscript lacks an essential description of material characterization and separation methods on the lab scale, and the presentation of quantitative outcomes of physical parameters, like density, size distribution, elemental compositions, and mineral phase compositions. Besides, presenting your experimental separation outcomes based on your estimation instead of real practice would be inappropriate.

Response 1:

With thanks, we appreciate your precise review. We tried to do our best to address all the revisions you requested. Please see our responses to your comments and the changes we have made to the manuscript in the following.

The introduction has been supplemented with sufficient background information and includes all relevant references. Research design was put in order and was reflected in all the necessary sections of the article. The research methods, as well as the equipment for conducting the study were given in a special section 2. The results of the study are presented in the corresponding tables (Table 1-4) and described in detail. The conclusion has been reworked.

The article was supplemented with a description of the characteristics of the material and methods of separation on a laboratory scale (Section 2.1), presentation of quantitative results of physical parameters such as density, size distribution, elemental composition and composition of the mineral phase. (Table 3). Moreover, the article presents the basic technological scheme of laboratory separation of the initial chrysotile-asbestos waste and its description (Figure 6).

Reviewer 2 Report

Authors have investigated the process of treating asbestos tailings and separating various kinds of concentrates. It is very meaningful. However, There are some points to be clarified as follow.

1. The abstract does not reflect the research content of this paper. It is suggested to rewrite the abstract.

2. Some data in this paper are too old. It is recommended to update some of the statistical data in the text, as shown in Figure 1.

3. Before proposing the flowchart of asbestos tailing processing in Figure 5, the data and analysis of the small-scale test should be described in detail. The description in the text is too brief.

4. The topic about asbestos waste treatment may interest the readers. However, economic evaluation should be made to ensure its application value.

5. Conclusion: please summarise the section with crisp findings with values.

6. Line 21: ‘comprised’ misspelling?

Author Response

Response to Reviewer 2. Comments

Comments and Suggestions for Authors

Authors have investigated the process of treating asbestos tailings and separating various kinds of concentrates. It is very meaningful. However, There are some points to be clarified as follow.

  1. The abstract does not reflect the research content of this paper. It is suggested to rewrite the abstract.

Response 1: 1.    Abstract of the paper was completely reworked. The abstract presents the idea of the article, as well as the main results.

  1. Some data in this paper are too old. It is recommended to update some of the statistical data in the text, as shown in Figure 1.

Response 2: 2.    The statistical data in the text, as shown in Figure 1 was updated.

  1. Before proposing the flowchart of asbestos tailing processing in Figure 5, the data and analysis of the small-scale test should be described in detail. The description in the text is too brief.

Response 3: 3.    The description of the laboratory scale tests was described in detail. The basic technological scheme, its description, as well as the results of the separation of raw materials into products was also described.

  1. The topic about asbestos waste treatment may interest the readers. However, economic evaluation should be made to ensure its application value.

Response 4: 4.    It was not possible to carry out an economic assessment on the treatment of asbestos waste.However, based on the fact that a variety of products can be obtained during processing, we can say that the process is cost-effective and reduces the technogenic load on the environment.

  1. Conclusion: please summarise the section with crisp findings with values.

Response 5: The conclusion was revised. It provides clear conclusions indicating the values.

  1. Line 21: ‘comprised’ misspelling?

Response 6: 1.    A spelling error in line 21 in the word "‘comprised’' has been corrected.

Round 2

Reviewer 1 Report

1. The structure of the article can be further adjusted. The detailed information on phase and elemental compositions of asbestos tailings in section 2.1 should be shifted to section 3 results and discussion. Section 2.1 should only include where and how were the samples collected.

2. Section 4 is missing between sections 3 and 5, probably due to the wrong numbering.

3. The authors should seek language proof-reading from a local English editor.

Author Response

Response to Reviewer 1. Comments

With thanks, we appreciate your precise review. We tried to do our best to address all the revisions you requested. Please see our responses to your comments and the changes we have made to the manuscript in the following.

  1. The abstract needs to be rewritten to be concise and conclusive. Remove unnecessary characterization results.

Response 1: 1. The abstract was redesigned according to your comments, unnecessary characterization results was removed.

  1. Figure 3. Is not clear, it should be change to a clear one. The X axis should add a unit.

Response 2: 2. Figure 3 was replaced with a clear one. Units of measurement have been added on the X-axis.

  1. What is the meaning of “fraction size -2.0+0.5, -0.5+0.25, -0.25”? Why do sizes have negative numbers?

Response 3: 3. Negative values indicated that the fraction size was less than two millimeters. Changes was made to table 3 for a clear understanding of the particle size.

  1. How is the gold content (0.3 g/t) determined?

Response 4: 4. The determination of gold was carried out according the method described in the following article:

Blyth, K. M., Phillips, D. N., & van Bronswijk, W. (2004). Analysis of Gold Ores by Fire Assay. Journal of Chemical Education, 81(12), 1780. https://pubs.acs.org/doi/10.1021/ed081p1780

  1. Page 5, line 172-174, a comma is followed by a lowercase letter.

Response 5: 5. This error has been fixed.

Reviewer 2 Report

1. The abstract needs to be rewritten to be concise and conclusive. Remove unnecessary characterization results.

2. Figure 3. Is not clear, it should be change to a clear one. The X axis should add a unit.

3. What is the meaning of “fraction size -2.0+0.5, -0.5+0.25, -0.25”? Why do sizes have negative numbers?

4. How is the gold content (0.3 g/t) determined?

5. Page 5, line 172-174, a comma is followed by a lowercase letter.

Author Response

Response to Reviewer 2. Comments

Comments and Suggestions for Authors

With thanks, we appreciate your precise review. We tried to do our best to address all the revisions you requested. Please see our responses to your comments and the changes we have made to the manuscript in the following.

  1. The structure of the article can be further adjusted. The detailed information on phase and elemental compositions of asbestos tailings in section 2.1 should be shifted to section 3 results and discussion. Section 2.1 should only include where and how were the samples collected.

Response 1: 1. The structure of the article was adjusted. The sections have been moved and supplemented according to your comments.

  1. Section 4 is missing between sections 3 and 5, probably due to the wrong numbering.

Response 2: 2. The sections of the article were numbered in the correct order.

  1. The authors should seek language proof-reading from a local English editor.

Response 3: 3. This article was previously checked by MDPI's English editing service. In addition, we made some changes to improve it.